# Characterization of Olive Oil Volatile Compounds after Elution through Selected Bleaching Materials—Gas Chromatography–Mass Spectrometry Analysis

**DOI:** 10.3390/molecules28186444

**Published:** 2023-09-05

**Authors:** Maher M. Al-Dabbas, Rawan Al-Jaloudi, Mai Adnan Abdullah, Mahmoud Abughoush

**Affiliations:** 1Department of Nutrition and Food Technology, The University of Jordan, Amman 11942, Jordan; m.aldabbas@ju.edu.jo; 2Science of Nutrition and Dietetics Program, College of Pharmacy, Al Ain University, Abu Dhabi P.O. Box 64141, United Arab Emirates; mahmoud.abughoush@aau.ac.ae; 3Department of Medical Science Support, Zarqa University College, Al-Balqa Applied University, As-Salt 19117, Jordan; rawanaljaloude@yahoo.com

**Keywords:** olive oil, oxidation, bleaching materials, volatiles, smoke point, partial refinery

## Abstract

Using different bleaching materials to eliminate or reduce organic volatiles in deteriorated olive oils will positively affect its characteristics. This study aims to identify the volatiles of oxidized olive oil after physical bleaching using selected immobilized adsorbents. Oxidized olive oil was eluted using open-column chromatography packed with silica gel, bentonite, resin, Arabic gum, and charcoal at a 1:5 eluent system (*w*/*v*, adsorbent: oxidized olive oil). The smoke point was determined. The collected distilled vapor was injected into GC-MS to identify the volatiles eluted after partial refining with each of these bleaching compounds. The results showed that volatile compounds were quantitatively and qualitatively affected by the type of adsorbents used for the elution of olive oil and the smoking points of eluted oils. The most prominent detected volatile compounds were limonene (14.53%), piperitone (10.35%), isopropyl-5-methyl-(2E)-hexenal (8.6%), methyl octadecenoate (6.57%), and citronellyl acetate (5.87%). Both bentonite and resin were superior in decreasing the ratio of volatile compounds compared with other bleaching materials used. Resin immobilized medium was significantly affected (*p* < 0.05), raising the smoke point. These results highlighted some information regarding the characteristics of volatile compounds that result after the physical elution of olive oil through selected adsorbents.

## 1. Introduction

Olive tree (*Olea europaea* L.) is widely cultivated in many parts of the world [1]. In Jordan, the production of olive fruits has increased significantly [2]. Recent studies have shown that the rate of olive fruit production between 2011 and 2020 was estimated to be about 154,000 tons, 78% of which was used as olive oil [3]. Olive oil is a vital agro-product to complement local food and is used in cooking, pharmaceuticals, cosmetics, medicine, fuel to light oil lamps, and soap production [4,5]. As a part of the Jordanian diet, it is consumed daily, with an average consumption of 4.6 kg per year [6,7].

Extra virgin olive oil (EVOO) is graded as the highest quality. It is extracted directly from olives at a low temperature through mechanical or physical methods [8,9,10]. EVOO has been widely associated with preventing cancer, heart disease, and aging by inhibiting oxidative stress [11] and diabetes mellitus type 2 [12]. A recent comprehensive systematic review and meta-analysis showed that around a 20% reduction in the relative risk of type 2 diabetes is associated with olive oil consumption [13]. These valuable properties are mainly attributed to its composition due to its high levels of monounsaturated fatty acids, particularly oleic acid, as well as containing natural antioxidants such as polyphenols and tocopherols [11,14]. Olive oil contains pigments, hydrocarbons, sterols, phospholipids, mono- and diglycerides, fatty alcohols, waxes, and diverse aroma compounds with relevant functions in olive oil stability and flavor [14,15]. Its composition depends on several parameters, including the cultivar, climate conditions, harvesting methods, fruit ripening degree, leaf removal, and crushing and extraction system [16].

Volatile compounds, defined as low molecular weight compounds with high vapor pressure at room temperature [17], are responsible for most of the sensory properties of olive oil and play a significant role in evaluating the overall oil quality, which has a decisive influence on acceptability [18]. They are mainly responsible for olfactory sensation, among other minor components [19]. However, the organoleptic characteristics and the quality of olive oil are adversely affected by unstable conditions over time during storage and transport, which alter the oil quality as a consequence of the degradation of some components [16,19,20,21]. Volatiles in virgin olive oil originate from three main well-known pathways [19]. The lipoxygenase pathway exerts the presence of C5 and C6 compounds that are responsible for the fruity aroma and sweet green/ripened perception of olive oil [22,23]. Volatiles such as nonanal, 2-pentyl furan, [19,22], C1–C4, and C7 are also formed due to the oxidation of fatty acids during olive oil storage, by which defective attributes will occur [23]. Bad practices during post-harvest or the storage of unfiltered oil may involve microbiological activities. Both oxidation and fermentation are attributed to the negative characteristics of olive oil [19,20,24].

A series of chemical reactions, such as oxidation, hydrolysis, and polymerization, takes place due to high temperatures and the presence of oxygen, moisture, and light, which result in the production of byproducts, including free fatty acids, hydroperoxides, alcohol, cyclic compounds, dimers, and polymers, which reduce oil shelf life and directly affect its quality [20,24]. It is worth mentioning that some traditional practices of old millers (in particular, in the northern part of Jordan) used to blanch the olive fruits in boiled water and followed by sun-drying for several days before being exposed to hydraulic pressing, causing highly oxidized oil known as “Saleeq” or “Maslooq oil”, which is classified as lampante virgin oil with undesirable organoleptic or chemical characteristics such as high acidity and peroxide that make it unfit for consumption according to IOC parameters [25]; however, some Jordanians believe in its benefits and prefer it. The blanching of olive fruit negatively affects the synthesis of volatile compounds and, thus, the aroma’s volatile profile [26]. Post-harvest practices such as keeping fruits in jute sacks for several days before milling and harsh methods in harvesting will lead to oxidation and fermentation reactions by exogenous enzymes that dramatically affect the volatile compounds and the final quality of the oil [17,27]. It has been reported that organoleptic defects are associated with the volatile composition of olive oil and are usually related to chemical oxidation [19,28].

Adsorbents have been used to refine crude oils to remove free fatty acids and other impurities that cause quality deterioration and reduce the oil’s shelf life. The partial refining of plant oils using adsorbents provides an environmentally friendly process that reduces the use of energy, toxic substances, and the loss of bioactive compounds while improving efficiency and product quality [29]. In previous work [30], selected immobilized adsorbents, silica gel, charcoal, activated bentonite, Arabic gum, and ion-exchange resin were used to partially refine oxidized olive oil and its quality was evaluated. To our knowledge, no studies have discussed the characteristics of volatile compounds reserved after oxidized olive oil’s physical elution through these selected adsorbents. Thus, the present work aimed to identify volatile compounds at smoke point temperature for olive oil after partially refining through five selected immobilized adsorbents (silica gel, bentonite, resin, Arabic gum, and charcoal) using GC-MS.

## 2. Results and Discussion

Lampante olive oil is intentionally refined to remove free fatty acids, peroxides, phospholipids, and volatile compounds in order to render it fit for human consumption [31]. It is well known that treatment with adsorbents will efficiently enhance the quality of used oil [29]. In the current study, olive oil was exposed to oxidation until achieving rancidity where the peroxide level was elevated to 48 mEqO_2_/Kg. Five adsorbent media were used to assess their efficiency in adsorbing organic volatiles accumulated in the oxidized and deteriorated olive oil. Volatiles were characterized after elution samples through selected adsorbents using GC-MS.

### 2.1. Effect of Adsorbents on the Volatile Compounds of Treated Olive Oil Samples

Volatile compounds are generally grouped as aldehydes, alcohols, esters, terpenes, and organic acids [32]. Table 1 shows the volatile compounds and their percentage quantity detected in the control sample where the oxidation of the oil was induced through direct exposure to a fan air current for a month. In this study, thirty-seven compounds were detected, namely, limonene (14.53%), piperitone (10.32%), 2-isopropyl-5-methyl-(2E)-hexenal (8.6%), methyl octadecenoate (6.57%), citronellyl acetate (5.87%), and 4,4-dimethyl-1-heptene-6-yne (5.18%), which were the most predominant compounds compared with others. All identified volatile compounds belong to several chemical classes, such as aliphatic hydrocarbons, terpenoids, esters, alcohols, furan, ether, heterocyclic compound, ketones, and aldehydes. It has been reported that when storing olive oil under the harsh conditions of air exposure, volatiles of penten-3-ol and hexanal will be predominant [20]. It is encouraging to compare the results of this current with that found by Romero et al. [33], who conducted a study aiming to validate a method of using solid-phase microextraction GC/MS in the detection of the volatiles that are responsible for negative organoleptic defects in virgin olive oil. The latter study stated twenty-nine of the volatile compounds responsible for the most organoleptic sensation in virgin olive oil. Among these are octane, ethyl acetate, ethanol, pentanal 3-pentanone, 1-penten-3-one, 2-butanol, ethyl butanoate, and hexanal. Another reported document showed that there were many volatiles detected in defective olive oil. Among these were E-2-hexen-1-ol, octane, hexanoic acid, ethyl acetate, propanoic acid, and 6-methyl-5-hepten-2-one [34]. While we shared some of these findings [20,33,34], the purpose, design, origin of the samples, sample preparation, and induced oxidation condition were different in the current study, and accordingly, a different volatile profile was obtained. Furthermore, the induced oxidation of our sample may alter the volatile compounds, causing further degradation.

It should be noted that due to the adsorption process, volatile compounds were quantitatively and qualitatively affected; not only were their detected levels decreased, but also the majority vanished to the point of no detection. Accordingly, these changes are often reflected in the smoke point. It was proven that the refinery process efficiently removed most of the volatiles [23].

The primary purpose of this paper is to explore the effect of selected adsorbent media on the efficient decreasing of volatile compounds accumulated after oxidation. The results of this study indicate that both the resin (40.18%) and the bentonite (42.31%) were superior in decreasing the number of volatile compounds by adhesion with adsorbents used to elute the oxidized oil (second part of Table 1). Moreover, the last section of Table 1 shows that the oil retained after elution through different adsorption media were almost the same, except that reserved through resin adsorbent, where the highest retention amount was detected (97.9%). Studies showed that silica adsorbent media efficiently reduced the amount of free fatty acids, aldehydes, phosphatidic compounds, and ketones [30,31]. The findings of this study showed that almost half of the volatiles were removed by silica compared to the control. It was reported that treating frying oil with a mixture of gel-derived alumina and activated clay or magnesium silicate significantly reduces the aldehydes, ketones, and odor [31]. Studies showed that bentonite is favored among other adsorbent media, such as silica and activated carbon, in bleaching vegetable oils due to its powerful adsorption capacity and affordable cost [35]. Moreover, the activation of bentonite increases its adsorptive properties and catalytic ion exchangeability due to chemical and mineralogical structure rearrangement [36]. This may explain the results obtained in this study, where bentonite was superior in removing volatiles compared with other adsorbents. It has been recorded that Arabic gum, which is a natural polysaccharide, acts as an emulsifier and has the ability to bind polar compounds through its OH groups by hydrogen bonding. Consequently, it could reduce some of the primary oxidative products in the eluted oxidized olive oil [37,38]. The results of this study indicate that Arabic gum has the ability to reduce almost half of the volatiles of oxidized oil. However, when comparing the efficiency of removing volatiles among these selected adsorbents, Arabic gum exhibited the lowest.

Out of the 37 volatile compounds detected in the oxidized olive oil of the control sample (Table 1), only three volatiles were found in the olive oil samples bleached through the five selected adsorbents: silica gel, bentonite, resin, Arabic gum, and charcoal using a 1:5 ratio of *w*/*v* adsorbent:oxidized olive oil. Those chemicals were limonene, ethyl sorbate, and thujone (Figure 1).

Four volatile compounds were detected in the oil eluted from four out of five adsorbent compounds: 4,5-dimethyl thiazole, 1-p-menthene, γ-terpinene, and trans-vertocitral C. Table 2 illustrates the percentage of these compounds and the medium used for elution. It was observed that 4,5-dimethyl thiazole, γ-terpinene, and trans-vertocitral C were not detected when Arabic gum was used as physical adsorbent media, while 1-ρ-menthene was not detected if elution was performed with bentonites.

On the other hand, 40% of the volatile compounds detected in the oxidized olive oil samples were removed after the elution. Those were *cis*-linalool oxide, p-menthane monoterpenoid, n-decanol, 1,1-dimethoxy-2-nonyne, citronellyl acetate, piperitenone oxide, myltayl-4(12)-ene, *trans*-cadinene ether, citronellyl pentanoate, 10,11-dihydroatlantone (E), caryophyllene acetate, eudesm-7(11)-en-4-ol, acetate, n-hexadecanol, methyl octadecenoate, and methyl octadecenoate. The removal of these constituents indicates the efficient reduction of furans, sesquiterpene, ketones, alcohols, and esters. Consequently, sensory defects will be reduced and the oil quality will improve.

Volatile compounds affect the aroma of olive oil [28]. Some of these are formed during the ripening process, whereas most are formed due to chemical and enzymatic reactions during the processing step. Enzymatic reactions, primarily through lipoxygenase, are responsible for forming volatile compounds and play a significant role in aroma evolution [16,28]. On the other hand, chemical reactions resulting from oxidative rancidity generally form unpleasant volatile compounds [39]. Aldehydes, ketones, esters, furans, aliphatic and aromatic hydrocarbons, and alcohols are the major volatile compounds that affect the quality of olive oil [40]. Their composition varies depending on the enzymatic activity [16,41]. On the other hand, factors such as cultivar, climate, ripeness, region, altitude, and processing conditions (washing, crushing, extraction processes, olive storage, harvest malaxing, and storage) also influence the volatile composition [32,39]. The level of volatile compounds is also correlated to the sensory quality [40]. To date, no attempts have been made to examine the volatiles of oxidized or lampante olive oil after bleaching with physical adsorbents. Overall, these results indicate that the selected adsorbents were efficiently able to remove these volatile compounds; hence, we can suggest executing physically partial bleaching through the adsorbent’s medium as a solution to remove volatiles that may affect the oil negatively.

### 2.2. Effect of Adsorbents on the Smoke Point of Treated Olive Oil Samples

The smoke points (°C) of the treated olive oil samples eluted by different adsorbents are presented in Table 3. The smoke point was significantly (*p* < 0.05) increased compared with that of the control sample (134.5 °C ± 0.71). Resin medium showed a superior increase in smoking point (188.5 °C ± 0.71) compared with silica gel (170.77 °C ± 0.42) > bentonite (160.50 °C ± 0.71) > Arabic gum (150 °C ± 1.41) ≈ charcoal (149.95 °C ± 0.07). All treatments significantly increased the smoke point of the oxidized olive oil after elution through the studied adsorbents. In this study, the improvement efficiency (%) ranged from 11.49% (for charcoal) to 40.15% (for resin).

The smoke point is the temperature at which a visible and continuous bluish smoke appears [42]. At this point, sufficient volatile compounds, such as free fatty acids and short-chain oxidation products, are emerging and evaporating from the oil. The smoke point of oils generally increases as the free fatty acid content decreases and the degree of refinement rises [43]. However, the smoke point should not be considered a reliable measure of an oil’s stability and suitability for cooking [44]. The current study employed a mixed-bed resin. It was reported that cation exchange resin effectively removes the free fatty acids by an esterification mechanism. In contrast, anion exchange resin can catalyze the transesterification of the tri-and diglycerides and remove fatty acid methyl esters. Moreover, the resin has adsorption ability and can adsorb water if contained in the oil [45]. The results of investigating the effect of adsorbent media on the smoking point of the eluted oil will now be compared to the findings observed when studying the impact of these adsorbents on volatile compounds.

It is apparent from the results obtained from Table 1 (% of volatiles removed, and % of oil retained) that resin and bentonite where the most effective adsorbents in terms of their ability to decrease the volatile compounds; nevertheless, resin proved its capability to reserve most of the oil, as well as to raise the smoke point effectively (40.15 °C). It can therefore be assumed that resin may serve as a promising improvement technology as an adsorbent medium.

## 3. Materials and Methods

### 3.1. Chemicals

Silica gel powder (60–200 mesh), activated charcoal granules, mixed-bed resin, and activated bentonite were purchased from LABCHEM chemicals (Zelienople, PA, USA). Methanol, hexane, and diethyl ether (HPLC-grade) were purchased from ASTM Co. (West Conshohocken, PA, USA). Iso-octane (biosolve (Chimie SARL., Saint-Quentin-Fallavier, France), Arabic gum, and other chemicals of reagent grade were purchased from local companies.

### 3.2. Preparation of Immobilized Adsorbents

Silica gel powder (60–200 mesh) was used separately to coat the glass beads (6 mm in diameter). The coating was accomplished using a carboxy methyl cellulose plasticizer followed by spreading a known weight of adsorbents and then rolling and sieving to give the resultant glass bead adsorbent. Then, the immobilized adsorbent was dried at 105 °C for 2 h. Activated charcoal granules, amorphous mixed-bed resin, amorphous Arabic gum, and white amorphous silica gel fine crystals (3–6 mm granule size) were used directly without coating.

### 3.3. Olive Oil Sample Preparation

Locally harvested and mechanically extracted olive oil was purchased from the local market in Jordan. Fifteen kilograms of the freshly produced olive oil was placed in an open glass container with a large surface area and left in open fan-circulated air at room temperature for one month to induce the oxidation process. Free fatty acids [46] and peroxide value [47] (PV) were measured periodically to achieve rancidity, with PV equaling 48 mEq O_2_/Kg oil, and the free fatty acid was determined to be 1.15% expressed as oleic acid.

### 3.4. Elution of Oxidized Olive Oil

Three hundred grams of oxidized olive oil were eluted through an open glass chromatography column (10 × 75 cm) loaded with 60 gm of each immobilized adsorbent separately at a ratio of 1:5 of immobilized adsorbents: oxidized olive oil. The eluted olive oil from each adsorbent was centrifuged for 5 min at 3000 rpm (HERMLE Z 206A, GmbH, Ottobrunn, Germany).

### 3.5. Determination and Identification of Volatile Compounds

The determination and identification of volatile compounds for the control and eluted oil samples were carried out after collecting the distilled volatiles at 200 °C, using GC-MS analysis, Varian Chrompack CP-3800 GC/MS/MS-200 (Saturn) and capillary column (30 m × 0.25 mm (i.d) and 0.25 µm film thickness) of DP-5 (5% diphenyl, 95% dimethyl polysiloxane). Helium gas was used as a carrier at a flow rate of 0.9 mL/min. The column temperature was kept at 60 °C for 1 min (isothermal) and then elevated to 300 °C at 3 °C/min. The temperature of MS was adjusted to 180 °C and operated in electron ionization mode (EI; 70 eV) [48]. The identification of volatiles was based on the built-in libraries (NIST Co and Wiley Co., San Francisco, CA, USA) and by comparing their calculated retention indices (RI) relative to (C8–C20) n-alkanes literature values measured with columns of identical polarity [49].

Quantitative analysis was conducted using a Hewlett-Packard HP-8590 gas chromatograph equipped with a split-splitless injector with a 1:50 split ratio and an FID detector was used. The column was an optima-5 fused silica capillary column (30 m × 0.25 mm, 0.25 μm film thickness) of 5% diphenyl, 95% dimethyl polysiloxan. The temperature of the oven was increased at a rate of 10 °C/min from 60 °C to 300 °C and then held constant at 300 °C for 5 min. The relative peak areas of the volatile components were measured and then used to calculate the concentration of the detected compounds. Each volatile percentage represents the fraction that contributed from total volatiles after elution through each adsorbent [48].

### 3.6. Smoke Point Determination

Smoke point is the temperature at which an oil begins to smoke continuously and can be noticed as bluish smoke [42]. The smoke point is an important parameter that indicates the event of chemical breakdown of the oil [44]. Smoking point apparatus was used to determine the smoke point for control and eluted samples following the AOCS standard method [42]. The oil is heated in cups, and the temperature is recorded as the smoke point when light blue smoke is noticed.

### 3.7. Statistical Analysis

A Randomized Complete Block Design (RCBD) was followed with blocking on replicates. The analysis of variance (ANOVA) of the smoke point data was carried out using Statistical Analysis System (SAS) program [50]. The results of the study were performed using ANOVA, and values were given as means ± standard deviation (SD). Least significant differences (LSD) at a 5% probability level were used to separate the means. All smoking point measures were conducted in triplicate. The results related to volatiles, their content, retained oil, and removed volatiles were presented through descriptive statistics.

## 4. Conclusions

This study set out to identify and characterize the volatiles of oxidized olive oil after eluting through silica gel, bentonite, resin, Arabic gum, and charcoal at a ratio of 1:5 eluant. The vapor was injected into GC-MS to identify these volatiles. The volatile compounds characterized in this study were quantitatively and qualitatively affected by the adsorption process. Furthermore, the evidence from this study suggests that the adsorbents used had a significant (*p* < 0.05) effect on increasing the smoke point of the olive oil. These findings enhance our understanding of the effect of the partial physical refinery of deteriorated olive oil on its volatiles, and will serve as a base for future studies to investigate the impact of using different adsorbents to decrease the volatile compounds in olive oil.

## Figures and Tables

**Figure 1 molecules-28-06444-f001:**
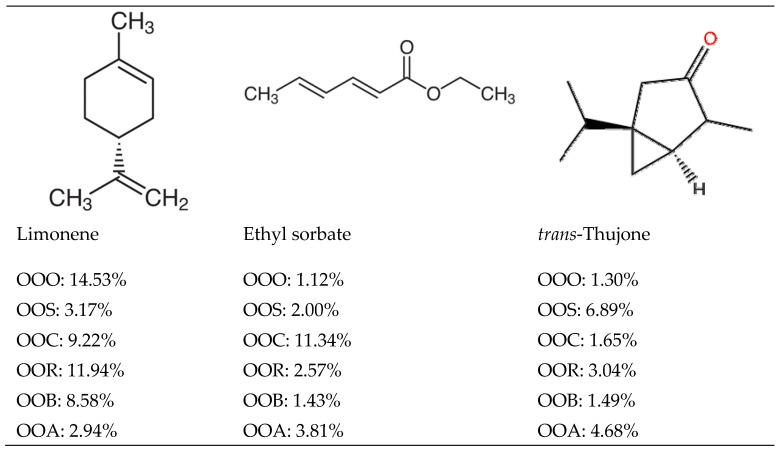
Chemical volatiles and their % detected after elution with different adsorbents. OOO: oxidized olive oil, OOS: oxidized oil eluted through silica gel, OOC: oxidized oil eluted through charcoal, OOR: oxidized oil eluted through resin, OOB: oxidized oil eluted through bentonite clay, and OOA: oxidized oil eluted through Arabic gum.

**Table 1 molecules-28-06444-t001:** Volatile compounds detected in oxidized olive oil (control), their %, chemical class, % of volatiles detected and % of oil retained after elution through different bleaching media compared with the control.

No.	Volatile				%	Chemical Class		
1.	Limonene ((−)-p-Mentha-1,8-diene, (−)-Carvene, (S)-4-Isopropenyl-1-methyl cyclohexene				14.53	Aliphatic hydrocarbon/a cyclic monoterpene		
2.	Piperitone				10.32	p-Menthane monoterpenoid/a cyclic terpene ketone		
3.	2- Isopropyl-5-methyl-(2E)-hexenal				8.6	Medium-chain aldehydes		
4.	Methyl octadecenoate				6.57	Hydrocarbon		
5.	Citronellyl acetate				5.87	Fatty acid ester/monoterpenoid		
6.	4,4-Dimethyl-1-heptene-6-yne (1-Hepten-6-yne)				5.18	Aliphatic hydrocarbon		
7.	β-Ocimene (E)				4.27	Hydrocarbon/monoterpenes		
8.	Citronellol isobutanoate				3.53	Fatty alcohol esters		
9.	n-Decanol				2.98	Straight-chain fatty alcohol		
10.	Valeranone				2.95	Ketone		
11.	1-ρ-Menthene				2.23	Menthane monoterpenoids		
12.	1-Octadecene				1.89	Alpha-olefin/long-chain hydrocarbon alkene		
13.	Caryophyllene acetate				1.86	Carboxylic acid Ester		
14.	Methyl cyclohexyl carboxylate				1.76	Ester		
15.	Dehydroxy_cis Linalool oxide				1.64	Acyclic monoterpene tertiary alcohol		
16.	1-Undecyne				1.64	Terminal acetylenic compound/alkyne		
17.	γ-Terpinene				1.62	Isomeric hydrocarbons/terpenes		
18.	5-neo-Cedranol				1.5	Alcohol		
19.	Myltayl-4 (12)-ene				1.4	Sesquiterpene		
20.	Eudesm-7(11)-en-4-ol, (acetate (7(11)-Selinen-4.alpha.-ol)				1.37	Sesquiterpenoids		
21.	Piperitenone oxide				1.33	Aliphatic heterocyclic oxepanes		
22.	*Trans*-Thujone				1.30	Monoterpene ketone		
23.	4,5-dimethyl Thiazole				1.28	2,4-disubstituted thiazoles		
24.	2-allyl-Phenol				1.27	1-hydroxy-4-unsubstituted benzenoids/phenols		
25.	n-Hexadecanol (hexadecanol)				1.17	Alcohol		
26.	Sandaracopimarinol				1.15	Terpenoids		
27.	Ethyl sorbate (Ethyl trans, trans-2,4-hexadienoate)				1.12	Fatty acid esters		
28.	1,1-dimethoxy-2-Nonyne				1.05	Acetal		
29.	Trans-Vertocitral C				1.01	Aldehyde		
30.	γ-Terpineol				1.01	p-menthane monoterpenoid		
31.	n-Pentadecanol				1.01	Alcohol		
32.	Citronellyl pentanoate (Citronellyl valerate)				0.98	Fatty alcohol esters		
33.	n-Pentadecane				0.96	Aliphatic alkane Hydrocarbon		
34.	10,11-Dihydroatlantone (E)				0.94	Sesquiterpenoid		
35.	*cis*-Linalool oxide				0.93	Tetrahydrofurans		
36.	trans-Cadinene ether				0.89	Ether		
37.	Methyl octadecenoate				0.89	Ester		
**% of volatiles detected through different bleaching media compared with the control**								
**OOO**		**OOS**	**OOC**		**OOR**	**OOB**		**OOA**
100%		54.06%	50.4%		40.18%	42.31%		58.71%
**% of oil retained after elution through different bleaching media compared with the control**								
100%		94.3%	94.9%		97.9%	93.7%		94.9%

OOO: oxidized olive oil, OOS: oxidized oil eluted through silica gel, OOC: oxidized oil eluted through charcoal, OOR: oxidized oil eluted through resin, OOB: oxidized oil eluted through bentonite clay, and OOA: oxidized oil eluted through Arabic gum.

**Table 2 molecules-28-06444-t002:** Volatile compounds and their percentage detected through four out of five adsorbent compounds.

Volatile	Chemical Structure	Silica	Charcoal	Resin	Bentonite	Arabic Gum
4,5-dimethyl Thiazole	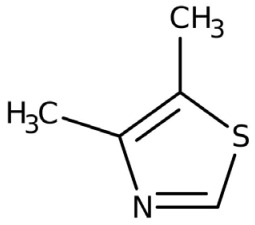	1.36	2.96	2.39	4.18	ND
1-ρ-Menthene	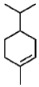	10.33	2.44	3.35	ND	6.39
γ-Terpinene	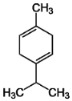	1.70	2.40	2.36	8.41	ND
Trans-Vertocitral C	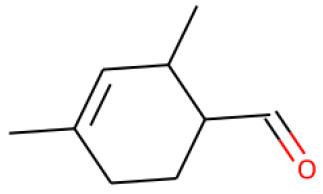	1.81	6.35	2.96	3.09	ND

ND: Not detected.

**Table 3 molecules-28-06444-t003:** Smoke point (°C) for oxidized olive oil (control) and after elution of oxidized olive oil through several adsorbents *.

Treatment	Smoke Point (°C) for Control Sample	Smoke Point (°C) for Treated Oil	Improvement Efficiency %
Resin	** 134.5 ± 0.71 ^e^	188.50 ± 0.71 ^a^	40.15
Charcoal		149.95 ± 0.07 ^d^	11.49
Arabic gum		150.00 ± 1.41 ^d^	11.52
Bentonite		160.50 ± 0.71 ^c^	19.33
Silica overnight		170.77 ± 0.42 ^b^	26.97

* Results are means of triplicate ± SD, and results with the same letter are not significantly different. ** this value was detected for control sample.

## Data Availability

Not applicable.

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
