# Peer review of "Characterization of Olive Oil Volatile Compounds after Elution through Selected Bleaching Materials—Gas Chromatography–Mass Spectrometry Analysis"

_molecules, 2023, doi:10.3390/molecules28186444_

Round 1

Author Response

I kindly present my comments about the article “Characterization of Olive Oil Volatile Compounds after Elution Through Selected Bleaching Materials – GC-MS Analysis”, I hope with this evaluation I succeed to establish a good reading and the Author of this document will be able to take the remarks in consideration.

Answer

Appreciate the time and effort of our esteemed reviewer; without his/ her input, the quality of this work would not be improved.

Moderate Editing of the English language was required.

Answer

Editing was conducted, and all typos were removed.

Abstract:

  • Please specify the 1:5 eluent system used.

Answer

Thank you for your comment.

1:5 eluent system: adsorbent: oxidized olive oil (w/v). It has been corrected within the context of the abstract and the methodology, and where required.

  • Note that oleic acid is not a volatile compound.

Answer

Thank you for your precious comment. Yes, you are right.

A Correction has been made.

Introduction:

  • Scientific names of species should be in italics.

Answer

Thank you for your precious comment.

A Correction has been made.

  • Capital letters should be used only at the beginning of sentences.

Answer

Thank you for your precious comment.

A Correction has been made.

  • Please review the sentence "Apart from being important from the sensorial point of view, the volatile fraction formed during the heating process is rich in degradation compounds." (Line 65 and 66).

Answer

Thank you for your precious comment.

A correction has been made.

  • Add a paragraph about the volatile composition of olive oil in the introduction.

Answer

Thank you for your precious comment.

A paragraph was added as per required.

Results & discussion:

  • Add the names of the adsorbents used. (Line 83).

Answer

Thank you for your precious comment.

Insertion of the adsorbents has been made.

  • Check the sentence "In general, aldehydes, alcohols, esters, terpenes, and organic acids are the main groups of compounds that volatiles consist of." (Line 87 and 88).

Answer

Thank you for your precious comment.

The sentence was revised and modification has been made.

  • Specify how the olive oil oxidation process was conducted. (Line 89).

Answer

Thank you for your comment.

In the methodology section, a full descriptive paragraph was available. Please see section “3.3 Olive oil sample preparation”. However, a short sentence was added as per your suggestion

  • Change "where" to "namely." (Line 90).

Answer

Thank you for your comment.

Modification has been made.

  • Capitalize the first letters of compound names only when they appear at the beginning of sentences.

Answer

Thank you for your comment.

All compound names were revised, and modification has been made.

  • Please correct (Line 90-92).

Answer

Thank you for your comment.

Revised, and modification has been made.

Note again that oleic acid is not a volatile compound.

Answer

I appreciate this precious comment. It was a mistake while transferring data. A correction was made as required. Thanks again.

  • For better readability, include the percentage next to their corresponding compounds.

Answer

Thank you for your comment.

Revised, and modification has been made.

  • Fatty acids are not volatile compounds.

Answer

Thank you for your precious and important comment.

Yes, for those with high C numbers, (C5, C6 are fatty acid volatiles).

  • Add a title to the table: "% of Volatiles Detected Through Different Bleaching Media Compared with the Control."

Answer

Thank you for your comment.

Correction has been made.

  • Authors mentioned in line 112 that only three compounds were detected in the treated oils compared to the untreated ones, then in line 120, they indicated the identification of four other molecules. Please clarify.

Answer

Thank you for your precious comment.

Three volatile compounds appeared in the five-bleaching media (All five-adsorbent media). While four volatiles were detected in four (out of five adsorbent media). I hope this was clear for you.  

  • There is no discussion of the finding results; authors are encouraged to discuss the importance of using adsorbents in improving the quality of olive oil and the mechanisms of adsorbing volatile compounds of each adsorbent used in the study.

Answer

Thank you for your precious comment.

Paragraphs were included.

  • Check the layout of table 2.

Answer

Thank you for your comment.

Sorry for such an error; this should be Table 2

  • On page 5, there is an issue with table numbering as two tables have been assigned the number 3.

Answer

Thank you for your precious comment.

Sorry for such an error; this should be Table 2

  • Consider removing "Table 3. Volatiles that were not Detected Through the Five Selected Adsorbents." It may not be necessary to indicate the non-detected compounds in a table.

Answer

Thank you for your comment.

Deletion has been made as per suggested.

  • The entry with two values of the smoking point for arabica gum is confusing. Please check table 4.

Answer

Thank you for your precious comment.

Sorry for the confusion; I made the suitable correction with a note.

  • Mention that Table 1 presents only the composition of the control sample. Check the sentence "It is apparent from the results obtained from Table (1) …" (Line 171).

Answer

Thank you for your precious comment.

A correction has been made, “adding: the second part of Table 1”, with an inclusion of a descriptive sentence for clarification” Thanks again.

  • To have a comprehensive view of the importance of using adsorbents in improving the quality of oxidized olive oil, authors should also determine the volatile composition of nonoxidized olive oil.

Answer

Thank you for your comment.

A paragraph was added.

Materials and Methods:

  • Mention the weight used for each adsorbent.

Answer

Thank you for your comment.

It was documented that 300g oil was used, and the ratio of the system was 1:5 w/v. A correction was made with the insertion of “60g” adsorbent.

  • Specify the quantity of oil collected in each treatment.

Answer

Thank you for your comment.

Insertion was made.

  • Clarify whether the authors developed the method used in volatile compounds determination or used a published method. In the latter case, please add the reference.

Answer

Thank you for your comment.

A reference was inserted.

  • Check and correct the sentence "Compounds identification was based on built-in libraries (NIST Co.; and Wiley Co., USA) by comparing retention indices (RI)" (Line 212).

Answer

Thank you for your precious and so important comment. A description of quantification results was missed due to an error while transferring the original data.

The paragraph was modified as: “The determination and identification of volatile compounds for control and eluted oil samples were carried out after collecting the distilled volatiles at 200 °C, using GC-MS analysis, Varian Chrompack CP-3800 GC/MS/MS-200 (Saturn) and capillary column (30m x 0.25 mm (i.d) and 0.25 µm film thickness) of DP-5 (5% Diphenyl, 95% Dimethyl Polysiloxane). Helium gas was used as a carrier at a flow rate of 0.9 ml/min. The column temperature was kept at 60 °C for 1 min (isothermal) and then elevated to 300 °C at 3°C/ min. The temperature of MS was adjusted to 180 °C and operated in electron ionization mode (EI; 70 eV). The identification of volatiles was based on the built-in libraries (NIST Co and Wiley Co, USA) and by comparing their calculated retention indices (RI) relative to (C8-C20) n-alkanes literature values measured with columns of identical polarity (Adams, 2017).

Quantitative analysis was conducted using a Hewlett-Packard HP-8590 gas chromatograph equipped with a split-splitless injector with a 1:50 split ratio and an FID detector was used. The column was an optima-5 fused silica capillary column (30 m × 0.25 mm, 0.25 μm film thickness) of 5% diphenyl, 95% dimethyl polysiloxan. The temperature of the oven was increased at a rate of 10 °C/min from 60 °C to 300 °C and then held constant at 300 °C for 5 min. The relative peak areas of the volatile components were measured and then used to calculate the concentration of the detected compounds. Each volatile percentage represents the fraction that contributed from total volatiles after elution through each adsorbent.

Additional reference is added: Adams, R.P. Identification of essential oil components by gas chromatography/ mass spectrometry, Ed. 4.1, 2017, ISBN 978-1-932633-21-4.

  • Explain how authors determined the relative percentage of compounds using a mass spectroscopy detector, as the MS detector does not consider the response factor. Usually, a FID detector is used to determine the relative percentage and calculate RI.

Answer

Thank you for your precious comment.

See above.

Reviewer 2 Report

Dear authors,

The manuscript entitled "Characterization of Olive Oil Volatile Compounds after Elution Through Selected Bleaching Materials – GC-MS Analysis" aimed to identify volatile compounds at smoke point temperature for olive oil after partially refining through different immobilized adsorbents using GC-MS. It presents scientific relevance for the area of Medicine, Chemistry and Food area.

After consulting www.sciencedirect.com and https://pubmed.ncbi.nlm.nih.gov/, publications were found for some authors involving the theme. However, you need to change some details/information in Introduction and discussion.

It is well written, but I suggest:

·         line 27 italic writing - (Olea europaea L.)

·         line 116 figure 1: free spaces in the notation of percentages

·         table 3 – please revise the name of the compounds

·         ND explicatie

·         the discussions should be further developed with an emphasis on variables.

·         pay attention to the degree Celsius symbol

·         line 308

·         line 338

·         references must be noted according to the guide

Author Response

After consulting www.sciencedirect.com and https://pubmed.ncbi.nlm.nih.gov/, publications were found for some authors involving the theme. However, you need to change some details/information in the Introduction and discussion.

I appreciate the time and efforts the reviewer made; this would definitely improve the quality of the work. On behalf of all authors, we thank you.

A modification was made.

It is well written, but I suggest:

  • line 27 italic writing - (Olea europaea L.)

Response:

Thank you for your comment.

A correction was made.

  • line 116 figure 1: free spaces in the notation of percentages

Response:

Thank you for your comment.

A correction was made.

  • table 3 – please revise the name of the compounds

Response:

Thank you for your comment.

A correction was made.

  • ND explicatie

Response:

Thank you for your comment.

An explanation was made as a footnote.

  • the discussions should be further developed with an emphasis on variables.

Response:

Thank you for your comment.

The discussion was improved.

  • pay attention to the degree Celsius symbol

Response:

Thank you for your comment.

A correction was made.

  • line 308

Response:

Thank you for your comment.

A correction was made.

  • line 338

Response:

Thank you for your comment.

A correction was made.

  • references must be noted according to the guide

Response:

Thank you for your comment.

A correction was made.

Reviewer 3 Report

The manuscript entitled:

“Characterization of Olive Oil Volatile Compounds after Elution Through Selected Bleaching Materials – GC-MS Analysis” describes a qualitative study of the volatile compounds in oils after physical bleaching using selected immobilized adsorbents.

In general, the work is good, and it could be of interest for the readers of the journal. Nevertheless, I have some remarks that could be taken into account to improve the manuscript.

My major comment regards the description of the outcome of the analysis, and are the following:

   1)     The authors should discuss more the results. In particular, they should articulate more the abundances of the various volatile compounds, and compare the outcome of their analysis with results obtained by other researchers. 

   2)     Please, make the results from the ANOVA more evident.

  3)     Are the % of volatile detected through different adsorbents comparable?

  4)     Table 3: Chemical structure of Menthene should be redrawn; it looks of a lower quality than the others.

  5)     There are several typos in the text, please, carefully re-read the manuscript and correct them.

  6)     “Volatiles that were not Detected Through the Five Selected Adsorbents” Please remove unnecessary capital letters

  7)     Line 228: “in triplicate. code.” Please correct

  8)     Line 257. “ is missing

  9)     Authors contributions are not described as required in the guidelines for the authors. Please reorganized them as required.

The quality of the English is fine, but there are several typos in the text 

Author Response

In general, the work is good, and it could be of interest for the readers of the journal. Nevertheless, I have some remarks that could be taken into account to improve the manuscript.

Thanks, on behalf of the authors, for the time and efforts executed to improve the work. Appreciated.

The English language was improved, as well as the result presentation.

My major comment regards the description of the outcome of the analysis, and are the following:

  • The authors should discuss more the results. In particular, they should articulate more the abundances of the various volatile compounds, and compare the outcome of their analysis with results obtained by other researchers. 

Answer:

Thank you for your precious comment.

Modifications and suitable insertion of required paragraphs were made.

  • Please, make the results from the ANOVA more evident.

Answer:

Thank you for your precious comment.

A modification was made. ANOVA was applied to the results of smoking points only. Statistical Results for smoking points were verified.

  • Are the % of volatile detected through different adsorbents comparable?

Answer:

Thank you for your precious comment.

A modification was made. NOT comparative! Each volatile percentage represents the fraction that contributed from total volatiles after elution through each adsorbent. So, it was impossible to compare with the same volatile in other adsorbents.

  • Table 3: Chemical structure of Menthene should be redrawn; it looks of a lower quality than the others.

Answer:

Thank you for your precious comment.

A modification was made as requested.

  • There are several typos in the text, please, carefully re-read the manuscript and correct them.

Answer:

Thank you for your precious comment.

A modification was made.

  • “Volatiles that were not Detected Through the Five Selected Adsorbents” Please remove unnecessary capital letters.

Answer:

Thank you for your precious comment.

A modification was made. This table was removed as per required by another reviewer.

  • Line 228: “in triplicate. code.” Please correct.

Answer:

Thank you for your precious comment.

A modification was made

  • Line 257. “ is missing.

Answer:

Thank you for your precious comment.

A modification was made.

  • Authors contributions are not described as required in the guidelines for the authors. Please reorganized them as required.

Answer:

Thank you for your precious comment.

A modification was made

Comments on the Quality of English Language: The quality of the English is fine, but there are several typos in the text 

Answer:

Thank you for your precious comment.

A modification was made